Genetic structure of marine and lake forms of Pacific herring Clupea pallasii

Orlova Svetlana Yu. kordicheva@rambler.ru kordicheva@vniro.ru 1 2
Rastorguev Sergey 3
Bagno Tatyana 3
Kurnosov Denis 4
Nedoluzhko Artem artem.nedoluzhko@nord.no 2 5
1 Russian Federal Research Institute of Fisheries and Oceanography , Moscow , Russia
2 Shirshov Institute of Oceanology of Russian Academy of Sciences , Moscow , Russia
3 National Research Center “Kurchatov Institute” , Moscow , Russia
4 Russian Federal Research Institute of Fisheries and Oceanography, Pacific Branch (TINRO) , Vladivostok , Russia
5 Nord University , Bodø , Norway
Jeffery Nicholas
Electronic publication date: 2021 Nov 2
Publication date: 2021
Volume: 9
Electronic Location ID: e12444
Received 2021 Mar 31; Accepted 2021 Oct 15
Copyright: ©2021 Orlova et al.
Copyright year: 2021
Copyright holder: Orlova et al.
License: This is an open access article distributed under the terms of the Creative Commons Attribution License, which permits unrestricted use, distribution, reproduction and adaptation in any medium and for any purpose provided that it is properly attributed. For attribution, the original author(s), title, publication source (PeerJ) and either DOI or URL of the article must be cited.
License URL: https://creativecommons.org/licenses/by/4.0/

Keywords: Pacific herring, Clupea pallasii, Speciation, Subspecies, Ecological form, RAD sequencing, Marine form, Freshwater form, Isolation, Russia

Funding: The Ministry of Science and Higher Education of Russian Federation 075-15-2021-1084 (contract: # 15.IP.21.0010) The Federal Agency for Fisheries under the State assignments No.076-00005-20-02, No.076-00005-19-01 RFBR (Russian Foundation for Basic Research) 19-54-54004 Nord University Open Access Fund covers the OA publication costs This work was supported by the Ministry of Science and Higher Education of Russian Federation, grant # 075-15-2021-1084 (contract: # 15.IP.21.0010). This research was conducted according to VNIRO calendar plan and funded by the Federal Agency for Fisheries under the State assignments No.076-00005-20-02, No.076-00005-19-01. Sergey Rastorguev was supported by the RFBR (Russian Foundation for Basic Research) Grant no 19-54-54004. Nord University Open Access Fund covers the OA publication costs. There was no additional external funding received for this study. The funders had no role in study design, data collection and analysis, decision to publish, or preparation of the manuscript.

==============================
The Pacific herring (Clupea pallasii) is one of the most important species in the commercial fisheries distributed in the North Pacific Ocean and the northeastern European seas. This teleost has marine and lake ecological forms a long its distribution in the Holarctic. However, the level of genetic differentiation between these two forms is not well known. In the present study, we used ddRAD-sequencing to genotype 54 specimens from twelve wild Pacific herring populations from the Kara Sea and the Russian part of the northwestern Pacific Ocean for unveiling the genetic structure of Pacific herring. We found that the Kara Sea population is significantly distinct from Pacific Ocean populations. It was demonstrated that lake populations of Pacific herring differ from one another as well as from marine specimens. Our results show that fresh and brackish water Pacific herring, which inhabit lakes, can be distinguished as a separate lake ecological form. Moreover, we demonstrate that each observed lake Pacific herring population has its own and unique genetic legacy.

Introduction

The Pacific herring (Clupea pallasii) is a teleost species from the Clupeidae family and is one of the most important fishery species. It inhabits cold waters of the Pacific and Arctic oceans. The natural distribution of the species covers (i) the North-Western Pacific from the Chukchi Sea to the Sea of Japan, and (ii) northeast European seas (White Sea, Pechersk Sea, and Kara Sea). Two ecological forms of Pacific herring (marine and lake) have been described to date (Andriyashev & Chernova, 1995; Gritsenko, 2002; Ivshina, 2001; Naumenko, 2001; Orlova et al., 2019). Approximately twenty lake Pacific herring populations were described in the North Pacific part of its distribution (Trofimov, 2002).

The lake ecological form of Pacific herring spawns and spends winter in brackish small bays and lagoon-type lakes having direct access to the sea without any long feeding migrations (Frolov, 1964), while the marine ecological form of herring spawns in large coastal bays, spends winter in the upper part of deep-sea on the seaward border of the continental shelf, and makes long feeding migrations (Naumenko, 2001).

Previous molecular-genetic studies based on analyses of allozymes, and mitochondrial and microsatellite DNA, suggest a separate genetic status of the marine and lake forms of Pacific herring (Gorbachev, Solovenchuk & Chernoivanova, 2011; Kurnosov & Orlova, 2021; Kurnosov, Orlova & Smirnova, 2016; Orlova et al., 2019; Rybnikova, Upryamov & Pushnikova, 1983; Semenova et al., 2012; Semenova et al., 2018; Shimizu, Takahashi & Takayanagi, 2018). However, the status of the lake form of Pacific herring is still under debate, since the differentiation between the forms was shown only on microsatellite markers (Orlova et al., 2019).

The restriction site-associated DNA sequencing (RAD sequencing) method (Baird et al., 2008) and its modifications, such as double digestion restriction site-associated DNA (ddRAD) sequencing (Peterson et al., 2012), have several advantages for large-scale evolutionary studies of non-model organisms, allowing to analyze the genomic variation of a significant number of individuals in a single sequencing run. These methods are usually used for genomics studies of non-model organisms focused on the analysis of genetic differentiation between natural populations to assess speciation (Franchini et al., 2017; Lal et al., 2016; Nedoluzhko et al., 2021; Nedoluzhko et al., 2020; Recknagel, Elmer & Meyer, 2013; Wessels et al., 2017).

In the present study, we analyzed the level of genomic diversity across wild populations of Pacific herring using the ddRAD sequencing method. Our results shed light on the genetic differentiation between lake and marine forms of this commercially important fish species. The molecular mechanisms underlying the origin of the new ecological forms in teleost species in different salinity conditions have been studied in detail (Raeymaekers et al., 2017; Rastorguev et al., 2017; Rastorguev et al., 2018; Terekhanova et al., 2019; Terekhanova et al., 2014), but key genomics and epigenomics drivers of this process remain unclear. Here, we describe genetic relationships among marine populations based on the polymorphism of a large number of SNPs by the ddRAD method and describe the population genomic structure of Pacific herring from the Kara Sea in the west to the Japan Sea in the south-east along its distribution.

Material and Methods

Sampling, DNA extraction, library preparation and sequencing

A total of 54 individuals from twelve wild populations of Pacific herring were collected from the Northwest Pacific and the northeastern European regions. Marine form individuals (except the Kara Sea population) were sampled during the spawning period. The individuals from Ainskoe Lake were also collected during the spawning period; the specimens from the lakes of Kamchatka (Bolshoy Vilyuy and Nerpiche lakes) were collected during the winter period. All samples were stored in the Russian Federal Research Institute of Fisheries and Oceanography (VNIRO), Moscow, Russia as ethanol fixed clips of fins. The number of specimens, population names, and their sources are shown in Table 1 and Fig. 1. The Kara Sea is a region where Pacific herring cohabit with Atlantic herring (C. harengus); therefore, specimens from this population were tested using mitochondrial DNA (mtDNA) control region (D-loop) to avoid species misidentification. Kara Sea specimens belonged to H2 and H7 Pacific herring haplotypes (Orlova et al., 2019).

Table 1 Pacific herring specimens that were used in this study and their sources.

#	Population abbreviation	Ecological form	Sampling location	Sampling period	Sampling coordinates (N; E)	Number of specimens	
1	Fvill	Lake	Bolshoy Vilyuy Lake	2016.02	52°49′25; 158°32′56	5	
2	Fain	Lake	Ainskoe Lake, Sakhalin Island	2010.06	48°29′44; 142°3′12	5	
3	Fnerp	Lake	Nerpiche Lake, Kamchatka	2008.03	56°22′28; 162°37′25	5	
4	Seve	Marine	Shelikhov Gulf, Sea of Okhotsk	2007.05	61°48′32; 159°22′52	5	
5	Sukur	Marine	Kuril Islands, Pacific Ocean	2017.06	44°14′55; 147°41′52	5	
6	Samur	Marine	Sea of Japan, Amur Bay	2017.04	43°4′23; 131°42′5	5	
7	Sk12	Marine	Bering Sea	2012.05	61°23′46; 179°10′11	5	
8	Skrg	Marine	Bering Sea, Karagin Bay	2007.06	58°36′6; 162°31′8	5	
9	Salex	Marine	Sea of Japan, Alexandrov Gulf	2009.05	50°57′0; 142°10′2	5	
10	Scva	Marine	Bering Sea, Gulf of Anadyr	2015.05	63°18′6; 175°26′4	3	
11	Sclu7	Marine	Sea of Okhotsk, Tugur Bay	2019.05	54°10′17; 136°59′4	3	
12	Scvk	Marine	Kara Sea	2014.10	70°24′; 65°37′	3	

Figure 1 Map showing the different locations in Russia where Pacific herring individuals were collected.

Sampling locations are indicated by red squares. Blue squares indicate lake populations, while red squares show marine populations.

Genomic DNA was isolated from ethanol-preserved fins using Wizard® SV 96 Genomic DNA Purification System (Promega, USA) according to the manufacturer’s recommendations. Purified DNA was quantified using a Qubit 2.0 fluorometer (Invitrogen, USA).

Fifty-four ddRAD libraries from the Pacific herring specimens (see Table 1) were constructed using the method described previously by Franchini et al. (2017), and using modifications described in Nedoluzhko et al. (2020). Genomic DNA was digested with MspI and PstI restriction endonucleases (NEB, Ipswich, USA). Amplified ddRAD libraries were quantified using a high-sensitivity chip on a 2100 Bioanalyser (Agilent, USA) and sequenced on the Illumina Novaseq6000 system (Illumina, USA) using paired-end reads (150 bp length). An S2 flow cell of Illumina Novaseq6000 genome analyzer (Illumina) with paired-end reads (2  × 150 bp length) was used for ddRAD library sequencing.

Bioinformatic analyses

Raw sequencing data were converted to FASTQ format with bcl2fastq2 (v2.20) and underwent a quality control check using FastQC (v0.11.5: https://www.bioinformatics.babraham.ac.uk/projects/fastqc/), and adapter removal using cutadapt (v2.10) with the quality parameter set to 30 (Marcel, 2011). Raw Illumina reads obtained from sequenced ddRAD libraries, were processed by Stacks package version 2.41 (Rochette & Catchen, 2017). The clone_filter module of the Stacks, was used for PCR duplicates removal. The process_radtags was used for demultiplexing the dual index reads, and for removing erroneous and low-quality reads (options: -c -q). Clean reads were mapped to the reference genome of Atlantic herring (Ch_v2.0.2) using Bowtie2 (Langmead & Salzberg, 2012) with the very-sensitive parameter. The mapped data in SAM format were converted to binary (BAM) format, sorted and indexed by Samtools (v1.7) (Li, 2011). SNP calling was conducted using Bcftools (v1.7) with –multiallelic-caller model and minimum base quality 30 (Li, 2011). The VCF file was additionally filtered by the genotyping quality parameter in the vcfR package (Knaus & Grunwald, 2017). Quality values were obtained using the getQUAL command. The loci with quality higher than 900 and with higher than 1000X coverage in all 54 Pacific herring individuals were used. The obtained VCF (filtered) file was loaded into the R statistical environment (v3.4.4) for discriminant analyses.

We estimated Nei’s genetic distances (Nei, 1978) to obtain the genetic divergence between populations, which varies from 0 (no common alleles) to 1 (all the frequencies of alleles are the same). The VCF file was then converted into genlight format of the adegenet R package (v2.1.3) (Jombart & Ahmed, 2011), and the StaMPP R package (v1.6.1) was used to calculate fixation index, Fst (AMOVA-based statistics) and Nei’s genetic distances (Pembleton, Cogan & Forster, 2013). A Fst was estimated by stamppFst() function of stampp package with 10,000 bootstrap replicas. We also used the adegenet R package for discriminant analysis (DAPC) to determine the number of clusters of genetically related individuals. Principal component analysis (PCA) was conducted using the glPca() function of the adegenet package (Jombart & Ahmed, 2011). The cluster tree topology based on genetic distances between populations was visualized in iTOL (v.4) (Letunic & Bork, 2019).

Results

DNA sequencing statistics and Pacific herring population structure

A total of 225,858,258 Illumina reads with a length of 150 nucleotides were produced from 54 Pacific herring specimens. The sequencing and mapping statistics for each DNA library are shown in Table S1. Initially, 1,909,252 of variable loci were found in 54 Pacific herring individuals (mean coverage –5 ×; mean missing data percent –60%). In total, 192,433 polymorphic loci (mean coverage –33. 6 ×; mean missing data percent –2.6%) for subsequent analyses remained after SNP calling and loci quality filtering.

Pairwise Fst genetic differentiation index was calculated for all studied Pacific herring populations (Table 2). P-value of the Fst estimation was equal zero for each pair population comparisons. The Fst distance between the Kara Sea population (Scvk) and other Pacific herring was the highest (ranging from 0.064 to 0.106), while the Pacific herring inhabiting north part of Bering Sea were more closely related to Kara Sea individuals (Fst = 0.05). The highest genetic distances were observed between individuals from the Kara Sea and three different lake forms, and this was independent of the geographical distance between locations. The PCA based on all SNPs also demonstrates these differences (Fig. 2A).

Table 2 Pairwise Fst values (marked by grey) and p-values of the comparison (above the main diagonal) for Pacific herring (Clupea pallasii) populations.

Population	Scva	Scvk	Fain	Fnerp	Fvill	Salex	Samur	Seve	Sk12	Skrg	Sukur	Sclu7	
Scva	–	<0.0001	<0.0001	<0.0001	<0.0001	<0.0001	<0.0001	<0.0001	<0.0001	<0.0001	<0.0001	<0.0001	
Scvk	0.050	–	<0.0001	<0.0001	<0.0001	<0.0001	<0.0001	<0.0001	<0.0001	<0.0001	<0.0001	<0.0001	
Fain	0.056	0.093	–	<0.0001	<0.0001	<0.0001	<0.0001	<0.0001	<0.0001	<0.0001	<0.0001	<0.0001	
Fnerp	0.046	0.079	0.040	–	<0.0001	<0.0001	<0.0001	<0.0001	<0.0001	<0.0001	<0.0001	<0.0001	
Fvill	0.077	0.106	0.051	0.029	–	<0.0001	<0.0001	<0.0001	<0.0001	<0.0001	<0.0001	<0.0001	
Salex	0.028	0.070	0.031	0.034	0.058	–	<0.0001	<0.0001	<0.0001	<0.0001	<0.0001	<0.0001	
Samur	0.023	0.079	0.044	0.043	0.064	0.015	–	<0.0001	<0.0001	<0.0001	<0.0001	<0.0001	
Seve	0.019	0.072	0.042	0.044	0.067	0.014	0.016	–	<0.0001	<0.0001	<0.0001	<0.0001	
Sk12	0.016	0.073	0.051	0.052	0.068	0.025	0.014	0.010	–	<0.0001	<0.0001	<0.0001	
Skrg	0.016	0.072	0.050	0.044	0.074	0.022	0.020	0.015	0.016	–	<0.0001	<0.0001	
Sukur	0.010	0.064	0.049	0.042	0.074	0.023	0.019	0.008	0.009	0.007	–	<0.0001	
Sclu7	0.036	0.082	0.063	0.057	0.085	0.034	0.032	0.028	0.028	0.025	0.032	–	

Figure 2 Genetic diversity between Pacific herring populations.

(A) Principal component analysis (PCA) plot by genotype distances for all Pacific herring populations. (B) PCA plot by genotype distances for Pacific herring populations from the Pacific Ocean part of its distribution. Blue area indicates lake populations, while marine populations are shown in red area.

Genetic differentiation between lake Pacific herring populations

Comparative analysis of the genetic differentiation between herring from lakes Bolshoy Vilyuy, Ainskoe, and Nerpiche on one side and combined marine population on the other, showed that lake form populations differ from the marine ones. The population from Bolshoy Vilyuy Lake had the highest Fst value (0.063) compared to combined marine individuals group, while the Fst value for Ainskoe and Nerpiche lakes was similar (0.041 and 0.040, respectively) (Table 2). The maximum differentiation was found between individuals from Ainskoe Lake (Sakhalin Island) and individuals from Bolshoy Vilyuy Lake and Nerpiche Lake (Kamchatka Peninsula). A lower level of differentiation was found between the two Kamchatka lakes (Fst = 0.029), although this value corresponds to the Fst values between Pacific herring samples from the most remote regions (Fst value between Amur Bay and the north Bering Sea populations = 0.023).

The PCA plot in Fig. 2 reflects the separate position of the majority of Pacific herring populations used in our study. Figure 2A shows all populations, including samples from the Kara Sea, which significantly differs from others. In Fig. 2B, for clarity, specimens from the Kara Sea are excluded from the plot, and marine and freshwater populations are highlighted by red and blue areas. Freshwater Pacific herring populations are clearly separated from marine ones.

DAPC analysis also revealed discrimination between Pacific herring populations from the Ainskoe, Bolshoy Vilyuy, and Nerpiche lakes and the Far Eastern seas (Fig. 3). Moreover, the Sakhalin population (Ainskoe Lake—Fain) differed significantly from the Eastern Kamchatka lake populations (Bolshoy Vilyuy—Fvil, Nerpiche—Fner).

Figure 3 DAPC plot (discriminant analysis of principal components) for Pacific herring populations from the Pacific Ocean part of its distribution based on individual genotypes.

Blue area indicates lake populations, while marine populations are shown in red area.

The genetic differentiation between lake populations was higher than that between marine populations, except for the Kara Sea. Large genetic differences were revealed for the lake populations—Nerpiche (Fst = 0.057), Ainskoe (Fst = 0.063), Bolshoy Vilyuy (Fst = 0.085)—compared to marine specimens, for which the Fst value ranged from 0.025 (Karagin Bay—Skrg) to 0.036 (Gulf of Anadyr—Scva). These results demonstrate that there is significant reproductive isolation between some Pacific herring lake forms and both marine and lake forms from other localities.

Genetic diversity between marine Pacific herring populations

Pacific herring from the Kara Sea population and lakes were excluded from the analysis for a detailed study of the population structure of the marine form of the Pacific herring. The average Fst value ranged from 0.007 to 0.036 for the Far Eastern marine populations of Pacific herring. The Tugur Bay population (Sclu7) from the Sea of Okhotsk was the most genetically distant population from the others (Fst index had the maximum value when compared with other marine populations—0.025–0.036).

The greatest genetic distance was found between Pacific herring individuals from the Gulf of Anadyr in the Bering Sea (Scva) and the Sea of Japan (Salex, Samur, Sukur). The individuals from the Gulf of Anadyr (Scva), Karagin Bay (Skrg), Bering Sea (Sk12), and Shelikhov Gulf of the Sea of Okhotsk (Seve) were genetically closely related. The individuals from the Gulf of Anadyr had the highest Fst value compared to all other marine populations. Populations from the Bering Sea were genetically close to the Shelikhov Gulf individuals, which presumably were progenitors for the Tugur Bay population (Sclu7).

An unrooted tree of Pacific herring populations was constructed based on the Nei’s distance matrix. The lake and the marine forms of Pacific herring were in separate clusters (Fig. 4).

Figure 4 Cluster analysis of Pacific herring performed on genome-wide identity based on Nei’s distances.

Blue font indicates lake populations, while marine populations are shown in red.

Discussion

Differentiation of the Pacific herring of the Arctic region

The sampling size in this study was limited by 3–5 specimens for one population, however, as was described previously the significance of the differences between the groups depends on both the number of specimens and the number of discriminating markers (Willing, Dreyer & Van Oosterhout, 2012). Test of significance for Fst results revealed all p-values were less than 0.0001 (Table 2).

We found a significant differentiation of the Kara Sea individuals from all other marine and lake Pacific herring populations. Our results suggest that the Arctic Ocean populations of this species were first to become separated from the others genetically. The lake populations of Sakhalin Island and the Kamchatka Peninsula were subsequently isolated. This was confirmed by the data from Ainskoe Lake (formed as lake 6–7.5 thousand years ago) (Budanov et al., 1957) and previously published studies that Pacific herring inhabited the Russian Arctic coastline more than ten thousand years ago (Grant, 1986). Our previous mtDNA study did not allow us to separate Arctic and Pacific Ocean populations of Pacific herring and the mass haplotypes were the same, but with a reduced value of haplotypic diversity in the Kara Sea population (Hd = 0.74) (Orlova et al., 2019).

A decrease in haplotypic diversity in the western part of the Russian Arctic has also been shown in comparison with the Pacific herring individuals of the Pacific Ocean, using a control region and cytochrome b gene sequence analysis (Laakkonen et al., 2013). The low genetic differentiation between individuals from the Kara and north Bering seas (Fst = 0.05), which was described in our study, may support the theory of the existence of a constant gene flow in historical terms, which may be increased in response to global climate change (Vermeij & Roopnarine, 2008).

Genetic diversity of the lake form of Pacific herring

Our results demonstrated the presence of the lake form of Pacific herring, which is genetically distinct from the marine form. By contrast, the previous measurements of morphological traits did not find any differences (Kartavtsev, Pushnikova & Rybnikova, 2008). Nevertheless, the results obtained in this study correspond to previous studies based on microsatellite and mtDNA markers (Kurnosov & Orlova, 2021; Orlova et al., 2019). Furthermore, Fst values for microsatellite data correlate with the same values obtained in this study. The lake forms from Ainskoe Lake and Kamchatka Lake have different origins, which form a separate lake clade (Fig. 4).

We assume that the mechanism of reproductive isolation between marine and lake Pacific herring populations is formed at the stage of spawning, and facilitates the conservation of the unique lake genotypes. Different optima of water salinity during the embryo development most likely provide reproductive isolation of lake forms of Pacific herring. Experiments on incubation of Pacific herring embryos from a lagoon-type lake showed their successful development at a salinity of 2.6‰ (Gritsenko, 2002), while embryos of the marine form died at a salinity below 4.65‰ (Fridlyand, 1951). Furthermore, it has been shown that embryos of Pacific herring from the Sea of Okhotsk died on spawning grounds where there were desalinated by freshwater (Tyurnin, 1965). Thus, our findings and those of previously published studies confirm the existence of a separate, genetically distinct lake form of Pacific herring (Fridlyand, 1951; Galkina, 1960; Gritsenko, 2002; Tyurnin, 1965). We assume the possible existence of strict natural selection as well as drift post divergence barrier for freshwater alleles, which occurs at the early embryo development of the lake form.

There are several studies that have described genome loci, or “genomic islands of divergence”, which are related to adaptation in plants and animals (Nosil, 2012). Previously, this type of natural selection of suitable freshwater alleles for these islands was considered as a way for the rapid local three-spined stickleback adaptation to low-salinity environments (Terekhanova et al., 2014). Three major loci with striking association to salinity have been described previously in the Atlantic herring genome. These potentially adaptive loci (prolactin receptor, high choriolytic enzyme, and solute carrier family 12 (sodium/chloride transporter) member 3) are related to osmoregulation and embryonal development (Martinez Barrio et al., 2016). Moreover, a 7.8-Mb inversion on Chromosome 12 as well as 125 loci associated with adaptation for the Baltic Sea conditions (including low salinity level) and 22 loci associated with different spawning times were described in the chromosome-level assembly of Atlantic herring genome (Pettersson et al., 2019). The positive selection has also been described in SYNE2, NRXN3B, CEP128, HK3 genes in both, Atlantic and Pacific herring species. Several SNPs in those genes have strong association with spawn timing (Petrou et al., 2021). At the same time, it is known that the spawn timing of Pacific herring, for example, in the Sea of Okhotsk, strongly depends on the ice situation and significantly depending on weather conditions from year to year. (Trofimov, 2006).

It is likely that the origin of the Pacific herring lake form followed the sticklebacks’ sympatric scenario (Terekhanova et al., 2019; Terekhanova et al., 2014). Sympatric speciation is usually explained by selection, where individuals/forms possess similar fitness, despite the different phenotypes. This type of selection often contributes to the expansion of a subset of phenotypic traits and the formation of evolutionary innovations (Hunt et al., 2011; Lahti et al., 2009; Snell-Rood et al., 2010). This type of shaping is shown in various teleost species such as barbs (De Graaf et al., 2010; Levin et al., 2020), Sevan trout (Levin et al., 2018), Arctic char (Alekseyev et al., 2002; Osinov, Volkov & Mugue, 2021), and cichlids (Barluenga et al., 2006; Kautt et al., 2020). At the same time we suggest that the geographic separation and glaciation during Pleistocene does not explain our findings, since the lake forms are genetically close to each other, but geographically distant. It seems that adaptive divergence in freshwater and brackish populations is more reliable (Orlova et al., 2019).

Analysis of genetic structure of the lake form of Pacific herring

Comparative genetic analysis of Pacific herring individuals from the Russian Far East showed that the Pacific herring population that inhabits Tugur Bay is the most genetically distinct from both marine and lake populations. The high level of genetic differentiation of this population relates to its strong isolation and may be associated with adaptation to the harsh conditions in this part of the Sea of Okhotsk.

Two large population groups can be distinguished in the North–West Pacific of the distribution of Pacific herring using genomic data: (1) Pacific herring of the Bering Sea, Karaginsky Bay, and the coast of the Kuril Islands; (2) Pacific herring of the Sea of Japan.

Such genetic differences can be explained by the partial isolation of the Sea of Japan and the presence of warm currents. The populations from the Sea of Japan are also distinct from each other. The Amur Bay population differs significantly from Sakhalin Island’s west coast population (Salex), while the marine population from Sakhalin Island have an intermediate position in their genetic structure between the Ainskoe Lake and Amur Bay populations (Fig. 3). This indicates the existence of partially separated subgroups of Pacific herring in the Sea of Japan.

Conclusions

The most detailed to date investigation of the genome-wide population structure of Pacific herring is presented here. The existence of a lake form of Pacific herring was confirmed based on 192,433 SNPs. Comparative analysis showed that the lake populations exhibit significant genetic differentiation from the marine form of Pacific herring. Moreover, a high level of genetic differentiation has been observed between different lake populations from Sakhalin Island and the Kamchatka Peninsula. The genomic data obtained in this study are in accordance with our results based on microsatellite and mtDNA markers, which confirms the formation of unique freshwater Pacific herring populations in lakes.

The Fst data obtained enabled us to divide the Pacific herring from the North-West Pacific part of its distribution into two large groups: The “North group”, which inhabits the Bering Sea, Kuril Islands, and the Karagin Bay; and the “South group”, which inhabits coastal waters of the Sea of Japan. We also found that the Sea of Japan populations significantly differ from each other based on Fst values.

Pacific herring from the Kara Sea differ significantly from the Pacific Ocean populations and lake populations (Kamchatka Peninsula and Sakhalin Islands). This indicates that the Kara Sea population became separated from the Pacific Ocean part of its distribution before the lake form of Pacific herring appeared.

Significant differences between all the studied populations, e.g., in comparing to the Atlantic herring populations on both sides of the North Atlantic Ocean (Fst ≤ 0.026) (Lamichhaney et al., 2017) suggest the existence of local populations of Pacific herring in each studied region. Interestingly, the Atlantic herring in contrast to Pacific herring spawns exclusively in deep waters, and seems lack reproductive isolation between fish stock in North Atlantic (Lamichhaney et al., 2017). In further studies, we will focus on a detailed description of these differences in frequencies of “freshwater” alleles that are responsible for the freshwater adaptation of Pacific herring populations, which inhabit lakes on Sakhalin Island and the Kamchatka Peninsula.

Supplemental Information

Supplemental Information 1 Illumina generated reads and number of mapped reads to Atlantic herring (Clupea harengus) reference genome sequence (Ch_v2.0.2)

Click here for additional data file.

The authors greatly appreciates to AV Orlyuk, and IK Trofimov from KamchatNIRO. The authors are also grateful to VV Gorbachev and SV Lipnyagov. The authors want to thank Prof. AM Orlov and Prof. BA Levin for their valuable comments, and Marchenko SL. for his help during Pacific herring sampling.

Additional Information and Declarations

Competing Interests

Author Contributions

DNA Deposition

Data Availability

The authors declare there are no competing interests.

Svetlana Yu. Orlova and Denis Kurnosov conceived and designed the experiments, performed the experiments, prepared figures and/or tables, authored or reviewed drafts of the paper, and approved the final draft.

Sergey Rastorguev analyzed the data, prepared figures and/or tables, and approved the final draft.

Tatyana Bagno conceived and designed the experiments, performed the experiments, analyzed the data, prepared figures and/or tables, and approved the final draft.

Artem Nedoluzhko analyzed the data, prepared figures and/or tables, authored or reviewed drafts of the paper, and approved the final draft.

The following information was supplied regarding the deposition of DNA sequences:

Data are available at NCBI SRA: PRJNA717668.

The following information was supplied regarding data availability:

The data is available at NCBI SRA: SAMN18506977–SAMN18507030.

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
