# Peer review of "Genetic structure of marine and lake forms of Pacific herring Clupea pallasii"

_PeerJ, doi:10.7717/peerj.12444_

## Round 0.1 · original submission · Major Revisions

Three reviewers provided helpful feedback and edits for this current draft of the manuscript. I agree that this is an interesting study that will be a useful contribution to this study system.

However, I share concerns with two of the reviewers that the sample size (n=5) for each population is very small. I agree that the significance of Fst estimates should be provided, as well as discussion on the potential effects of this small sample size on any inferences made. Additional details are needed in the methods as well, including all bioinformatics steps and details, as well as information on all analyses performed.

Please see comments from each reviewer and address these as part of your revised manuscript.

·

Basic reporting

no comment

Experimental design

no comment

Validity of the findings

no comment

Additional comments

This is good article that sheds light on herring freshwater ecological forms. Appropriate biological experiments and bioinformatics analysis were used and all findings were supported by relevant statistics results. I believe this article should be accepted as it is.

Reviewer 2 ·

Basic reporting

Some minor issues use of appropriate references, some details on filtering of data unclear

Experimental design

Some methodological details unclear, concern about inadequate sample size for analyses used and no tests of significance

Validity of the findings

Some issues with overinterpretation of observed patterns of population structure, difficult to assess validity of findings fully without greater communication of significance/confidence intervals

Additional comments

The authors present an investigation of population structure in marine and lake forms of Pacific Herring across the Northwest Pacific, using a large panel of ddRAD loci. They identify moderate population structure in their dataset, suggesting regional splits in populations of this species, supported by previous analyses using smaller panels of molecular markers. I found these results to be interesting, and likely good candidates for further exploration using larger groups of genotyped individuals, using either RAD sequencing or whole genome resequencing. However, I have some concerns about this manuscript in its current form:

1. The present results are based on limited number of individuals (54, - 4 to 5 per population). This study appears to use primarily population genetics tools (FST, PCA, DAPC), in a phylogeographic framework (sequencing a limited number of individuals from representative populations). This combination of approaches can limit the inferences possible from population genetic data – it is not clear if any of the observed patterns of differentiation are actually significant. I suggest presenting results from differentiation analyses more fully, with significance of assessed FSTs (pvalules) and confidence intervals of these values, which can be provided by StAMPP (stamppfst function). Similarily, the DAPC results indicate population structure, but the number of inferred clusters, which clusters are supported by information criteria, and whether the membership there matches geographic clustering, should also be estimated. Last, an AMOVA approach was used for estimating FST – I would suggest using this method and including hierarchical information based on geography or expected lineage to determine the proportion of variance attributable to some level of geographic or evolutionary separation.
2. I think emphasis on any adaptive component of speciation or population structure cannot be adequately addressed given the data. Geographic separation and glacial history also account for separation between populations, and should be included in the discussion.
Minor comments below:
Line 68: This is not the correct reference here – Baird 2008 https://journals.plos.org/plosone/article?id=10.1371/journal.pone.0003376
Line 70: ddRAD reference is Peterson 2012. - https://journals.plos.org/plosone/article?id=10.1371/journal.pone.0037135
Line 71: These methods have not reduced the cost of sequencing, but have made available surveys of genomic variation in non-model organisms. Suggest revising.
Line 72: no just aquatic, or animals. Suggest revising to “non-model organisms”
Line 102: revise as Franchini et al. (2017)

Line 114 : 117 : No mention of removal of PCR duplicates here – for ddRAD this can be challenging as there is not secondary shear site to suggest identical reads. Maybe additional filtering based on HWE deviation (potentially challenging with low sample number) or depth cutoffs (remove high depth sites, maybe beyond a given threshold e.g greater than 3sd mean coverage.)
Line 117: Suggest clarifying – more analyses than DAPC were carried out
Line 118: Reference needed here

Reviewer 3 ·

Basic reporting

Basic reporting:

The manuscript by Orlova et al. investigates the population structure of different ecotypes (lake vs. marine) of Pacific herring using ddRAD sequencing data. The authors find evidence of genetic divergence between ecotypes and between herring spawning in the Kara Sea. I enjoyed learning about this very interesting study system! Overall, professional English is used throughout, raw data have been shared on NCBI, and sufficient context for the research in included. The paper would be further strengthened if some of the some of the following issues are addressed:

1. The terms “freshwater” and “lake” herring are used interchangeably to describe herring that spawn and overwinter in small bays and brackish lagoons (line 59). This terminology is a bit confusing for the reader; for example, in my first reading of the paper I was under the impression that the lake ecotypes were in landlocked freshwater lakes and it was only until I saw Figure 1 that I realized that the “lake” ecotypes were collected from coastal Kamchatka and the Sea of Japan. It would be helpful if there was a short description of the “lake” vs. marine study sites in the methods section to help the reader visualize and understand the differences between these environments. If you have photos of the lake environments, it might also be good to include them as a figure (or in the supplement) to further illustrate this point. If there are no large differences between the lake and marine environments (or if they form a continuous gradient of habitat rather than a hard barrier to migration), then I might even suggest using different terminology to identify the two environments and ecotypes. The following terms could be used instead: lagoon (or low-salinity or resident or brackish) vs. migratory herring. Additionally, I recommend using different symbols in the figures (map, DAPC) to differentiate between the two herring ecotypes.

2. In Table 1 please provide the date of sampling at each location. Also in the methods section, clarify whether these fish were collected from active spawning events. Might differences in spawn timing also contribute to the deep genetic divergence of lake and marine herring? Here is a paper that explores the role of temporal isolation in Pacific herring that perhaps might be useful:

Petrou EL, Fuentes-Pardo AP, Rogers LA, Orobko M, Tarpey C, Jiménez-Hidalgo I, Moss ML, Yang D, Pitcher TJ, Sandell T, et al. 2021. Functional genetic diversity in an exploited marine species and its relevance to fisheries management. Proceedings of the Royal Society B: Biological Sciences 288:20202398.

3. Line 81: There is some really nice genomic research on adaptation to low salinity environments in Baltic Sea herring. You should cite that body of research in the introduction and/or in the discussion to provide a richer context for your work. Here are some key papers:

Martinez Barrio A, Lamichhaney S, Fan G, Rafati N, Pettersson M, Zhang H, Dainat J, Ekman D, Höppner M, Jern P, et al. 2016. The genetic basis for ecological adaptation of the Atlantic herring revealed by genome sequencing. eLife 5:e12081.

Pettersson ME, Rochus CM, Han F, Chen J, Hill J, Wallerman O, Fan G, Hong X, Xu Q, Zhang H, et al. 2019. A chromosome-level assembly of the Atlantic herring genome—detection of a supergene and other signals of selection. Genome Research.

4. In the results section the subheaders (3.1, 3.2, etc.) do not always correspond to the content below them. I recommend removing or renaming the subheaders and re-organizing the results section to improve flow. The sections reporting pairwise population FST could be significantly shortened and the results would be better reported in a table in the main manuscript (rather than as Supplemental Table 2).

Experimental design

Experimental design:
This article is within the aims and scope of the journal and the research question is well-defined. Please see the following comments for suggestions for improvement:
1. Line 67: I am not sure I understand what you mean by “ the status of lake form herring is still under debate”? Is that because previous studies used relatively few DNA markers? It would be good to clarify this point a little bit more, and I think it will help with the transition to the next paragraph.
2. Line 102: please name the digestion enzymes you used in ddRAD library preparation. I also suggest re-writing this sentence as: “… constructed using the method described by Franchini et al. (2017), and using modifications described in Nedoluzhko et al. (2020).
3. Line 105: this sentence about the bioanalyzer chips should come before the previous sentence about the S2 flow cell.
4. Line 115: Which variant calling model did you use in bcftools? (ex: --consensus-caller)
6. Line 142-144: The way this sentence is written makes me think that both lake AND marine herring were collected from Bloshoy Vilyuy, Ainskoe, and Nerpiche. But I don’t think that is the case? Please check this sentence for meaning.
7. In the results section, please provide some information about the average read depth per SNP, percent missing data per sample, percent missing data per SNP, and the number of SNPs with a global minor allele frequency above 5% (or thereabouts). I imagine that sequencing samples on an S2 flow cell results in low amounts of missing data but it would be good to report these metrics. If you have a vcf file, it is easy to produce these summary statistics using the program VCFtools.
8. Line 146: You should point the reader to Supplementary Table 2 here. Also this table caption has a typo in the species name (should be Clupea pallasii), and some of the population names do not match the Figure 1 map.
9. Line 163: Are these pairwise FST values? Clarify to what marine specimens they are being compared.
10. Line 191: Can you clarify what the data from Ainskoe Lake were? Were those dated ancient herring bones?
11. Line 199: Are the Chukchi Sea samples Scva and Sk12? I recommend labelling the Chukchi Sea in Figure 1 (map).
12. Line 220: This is an interesting section!
13. Line 222: The standard term is “genomic islands of divergence”
14. Line 225: “Fast evolution constructor from precast bricks” is a bit awkward. It would be better to say something like: “standing genetic variation enabled the rapid local adaptation to low-salinity environments”.
15. Figure 2A and 2B: Would it be possible to keep the color scheme consistent for each population between these two plots and also plot the lake herring with a different character? That would help showcase the differentiation between lake and marine ecotypes.

Validity of the findings

Validity of the findings
The number of specimens collected from each sampling location is very small (N=5). I think it would be good to address the fact that very small sample sizes are used and there is likely sampling error around estimates of population-specific allele frequencies and FST.

---

## Round 0.2 · Minor Revisions

The revised version of this manuscript addresses much of the concern that was discussed during the first review, including adding tests for significance and addressing the small sample size of the study. Some minor concerns remain, concerning adding references on the use of small sample sizes versus large numbers of loci, and additional details to the methods, which I agree need addressing prior to acceptance. One reviewer correctly notes that you do not test for genes associated with freshwater habitats, and so this should be removed from the introduction. Finally, some changes to the figures concerning the colour scheme and legend are required. I recommend addressing each of the reviewers' comments, which, while generally minor, will greatly improve the final draft of this manuscript.

Reviewer 2 ·

Basic reporting

No major issues here

Experimental design

Design appropriate, limited in sample size, but does not preclude accurate results meriting follow-up with larger datasets.

Validity of the findings

As above: small sample size, but large marker panel, and correspondence to previous datasets.

Additional comments

The authors present a revised manuscript exploring population structure in Pacific Herring populations. They have made several corrections and clarified sections to address some reviewer comments, but I still find some sections require additional minor revisions, outlined below:

In addressing the sample size limitations of the present study in their response, the authors assert that when sample sizes of markers are sufficiently large, a small number of sequenced individuals (n =4 – 6) maybe be sufficient. Some mention of this limitation, and discussion and an appropriate reference are still required here (e.g. https://doi.org/10.1371/journal.pone.0042649).

Section titles: Genetic diversity vs differentiation. This study relies on differentiation metrics (PCA, DAPC, FST), rather than diversity metrics (pi, heterozygosity). Correcting section titles is suggested.

Line 154: Which genotyping quality parameter and value was specified here? Depth or other?

Line 227: Unless a linear model was used and produced a specific correlation coefficient that was significant, I’d pick a less statistical term like: “is similar to”, “matches”, or “corresponds to”

Line 237: Genetically distinct?

Line 239: Not sure it needs to be strict natural selection, drift post divergence barrier may be implicated too

Line 242: I’d maybe drop this line, or move to a more general reference (e.g. Nosil 2012 Ecological Speciation book), these “islands” are identified across many species (e.g. sunflowers, stick insect, littorinid snails).

Line 260: Sympatric speciation is not generally associated with weak selection, the strength of selection needs to exceed rate of migration. I’d remove “weak” here.

Line 287: Low sample size (4-6 individuals) and RAD data (incomplete sampling) precludes a full assessment, rewording suggested here (e.g. “the most detailed genomic investigation to date”)

Reviewer 3 ·

Basic reporting

There are several errors in the figures and text that should be addressed before publication:

• In the downloaded materials, I did not see any figure captions. Are these missing from the manuscript?

• In the abstract and the first paragraph of the introduction, the authors should add that the species range of Pacific herring also extends to the Northeastern Pacific Ocean (there are a multitude of spawning locations in North America).

• In all of the Figures: In Figure 1, the Lake populations are depicted in blue and the marine populations are depicted in red. In figures 2 +3, the ellipses that encompass these ecotypes have the opposite color scheme, causing confusion for the reader. Please adjust figures so they have consistent color scheme.


• Figures 2 A and B: There is an error in the color scheme and the legend. In Figure 2A, I think that the Kara Sea individuals are in the upper right-hand corner, but this color is missing from the legend in the right. Additionally, according to the methods section there should not be Kara Sea individuals in Figure 2B, but the legend makes it seem like there are. Please fix these labeling errors.


Line 146 (tracked changes document): should be "bootstrap replicates"

Lines 183 - 185 (describing the results of the PCA): these lines do not actually convey any PCA results as written. Re-write so you describe the major patterns observed in the PCA.

Line 186: This should be in the methods section. Remove it from the results.


Line 192 : I don’t understand how the FST values are reported here. Clarify that these are the maximum pairwise FST values observed between a particular lake and marine population, and direct readers to Table 2.

Experimental design

The experimental design was adequate for the questions asked. But there is one thing that should be changed in the introduction:

Line 87: You do not test whether any genes are associated with low-salinity environments in this study, so you should remove the line where you state that your objective is to describe genes that are involved in adaptation to freshwater.

Validity of the findings

no comment

Additional comments

In the abstract and the first paragraph of the introduction, the authors should add that the species range of Pacific herring also extends to the Northeastern Pacific Ocean (there are a multitude of spawning locations in North America).

Line 87: You do not test whether any genes are associated with low-salinity environments in this study, so I would remove the line where you state that your objective is to describe genes that are involved in adaptation to freshwater.

Line 146: should read bootstrap replicates

Lines 183 - 185 (describing the results of the PCA): these lines do not actually convey any PCA results as written. Re-write so you describe the major patterns observed in the PCA.

Line 186: This should be in the methods section. Remove it from the results.

---

## Round 0.3 · accepted · Accept

The authors have done an excellent job of addressing the reviewers' comments over two rounds of revisions. This paper can now be accepted for publication, and thank both the authors and reviewers for their thoughtful and professional comments and critiques.